# Diagnostic and Therapeutic Pathway in Diffuse Malignant Peritoneal Mesothelioma

**DOI:** 10.3390/cancers15030662

**Published:** 2023-01-21

**Authors:** Shigeki Kusamura, Dario Baratti, Michele De Simone, Enrico Maria Pasqual, Luca Ansaloni, Daniele Marrelli, Manuela Robella, Fabio Accarpio, Mario Valle, Stefano Scaringi, Daniele Biacchi, Carmen Palopoli, Sergio Gazzanelli, Marcello Guaglio, Marcello Deraco

**Affiliations:** 1Peritoneal Surface Malignancies Unit, Fondazione Istituto Nazionale dei Tumori IRCCS Milano, 20133 Milan, Italy; 2Candiolo Cancer Institute, FPO—IRCCS, 10060 Candiolo, Italy; 3AOUD Center Advanced Surgical Oncology, DAME University of Udine, 33100 Udine, Italy; 4Unit of General Surgery, San Matteo Hospital, 27100 Pavia, Italy; 5Unit of General Surgery and Surgical Oncology, Department of Medicine, Surgery and Neurosciences, University of Siena, 53100 Siena, Italy; 6CRS and HIPEC Unit, Pietro Valdoni, Umberto I Policlinico di Roma, 00161 Rome, Italy; 7Peritoneal Tumours Unit, IRCCS Regina Elena National Cancer Institute, 00144 Rome, Italy; 8IBD Unit, DEA AOU Careggi, 50134 Florence, Italy; 9U.O.C.—P.S.G. con O.B.I. Azienda Ospedaliera Universitaria “G. Martino”, 98125 Messina, Italy

**Keywords:** diffuse malignant peritoneal mesothelioma, cytoreductive surgery, HIPEC

## Abstract

**Simple Summary:**

Mesothelioma is a rare cancer that originates in mesothelial surfaces of the peritoneum, pleura, and other sites. Diffuse malignant peritoneal mesothelioma (DMPM) corresponds to approximately 15% of all mesotheliomas. Cytoreductive surgery and HIPEC is the current standard of care, allowing the achievement of a median OS of at least 4 years. However, patient selection and perioperative management remain challenging. This paper outlines the diagnostic and therapeutic pathways of DMPM.

**Abstract:**

Diffuse malignant peritoneal mesothelioma (DMPM) is a rare form of mesothelioma that carries a very poor prognosis. The 5-year overall survival is about 20% (±5.9). Survival is optimal for patients suitable for cytoreductive surgery (CRS) with Hyperthermic Intraperitoneal Chemotherapy (HIPEC), with a median OS ranging from 34 to 92 months. However, selecting patients for surgery remains a complex task and requires a careful preoperative workup, rational analysis of prognostic profiles, and risk prediction models. Systemic chemotherapy could be offered: (1) in the adjuvant setting for high-risk patients; (2) for patients not eligible for CRS; and (3) for those with recurrent disease. It mainly includes the combination of Platin compound with Pemetrexed or immunotherapy. The biology of DMPM is still largely unknown. However, progress has been made on some fronts, such as telomere maintenance mechanisms, deregulation of apoptosis, tyrosine kinase pathways, and mutation of BRCA1-associated protein 1 (BAP1). Future perspectives should include translational research to improve our understanding of the disease biology to identify druggable targets. We should also clear the role of immune checkpoint inhibitors and investigate new locoregional technologies, such as pressurized intraperitoneal aerosol chemotherapy (PIPAC) or normothermic intraperitoneal chemotherapy (NIPEC).

## 1. Introduction

Mesothelioma is a rare cancer that originates in mesothelial surfaces of the peritoneum, pleura, and other sites. However, it can also occur rarely in other compartments, such as the pericardium and tunica vaginalis testis [1,2]. Diffuse malignant peritoneal mesothelioma (DMPM) corresponds to approximately 15% of all mesotheliomas. The United States has intermediate incidences ranging from 0.41 to 1.94 per 100,000 [3,4,5,6]. The highest rates are reported in the UK and Oceania, while some of the lowest reported rates are in Japan and central Europe. According to a population-based study from the SEER database, 1-year overall survival was estimated as 46% (±1.3) in patients with DMPM, and 5-year overall survival as 20% (±5.9) [7]. Survival is optimal for patients suitable for cytoreductive surgery (CRS) with intraperitoneal chemotherapy [8,9,10,11,12]. Several systemic chemotherapy options are available for patients who are not candidates for CRS or those whose disease recurs following surgical treatment. The most common is the combination of Platin compound with Pemetrexed. Similar to malignant pleural mesothelioma, the histologic subtypes of DMPM include epithelioid (most common), sarcomatoid, and biphasic [13]. Patients with epithelioid histology have better outcomes than those with either biphasic or sarcomatoid histologies. Although there are similarities between DMPM and its pleura counterpart, there are unique differences. DMPM is diagnosed in equal numbers of males and females, while pleural mesothelioma is more common in males [7]. In addition, DMPM may occur in younger patients, whereas pleural mesothelioma typically occurs in older patients. In contrast to its pleural counterpart, DMPM is less frequently associated with asbestos exposure [14,15]. In fact, only about 33–50% of patients diagnosed with DMPM report any known prior exposure to asbestos [4,16].

The biology of DMPM is still largely unknown, but progress has been made on some fronts. Telomere maintenance mechanisms, which account for many malignancies’ limitless cell replicative potential, are associated with the DMPM’s prognosis [17]. The dysregulation of the apoptotic pathways and its effectors (survivin, IAP-1, IAP-2, and X-IAP, Smac/DIABLO) provided some insights regarding the relative chemo-resistance of DMPM; it allowed the identification of novel targeted therapeutic strategies [18]. Other potential druggable targets were disclosed by investigating the dysregulation of the RTK signaling pathway involving EGFR, PDGFRA, and PDGFRB [19]. More recently, other insights were achieved regarding the role of microRNAs in DMPM pathophysiology, particularly miR-550a-3p [20], miR-34a [21], and miR-3805-p [22]. The epithelial–mesenchymal and mesenchymal–epithelial reverse transitions may be associated with the ability of epithelioid DMPM cells to spread peritoneally and promote drug-resistant phenotypes [23].

Other genetic factors play a critical role in the pathophysiology of some patients with DMPM. BRCA1-associated protein 1 (*BAP1*) is a deubiquitinating hydrolase that plays a crucial role in various cellular processes. Germline and somatic inactivation of BAP1 is frequent in mesotheliomas [24].

Patients with DMPM present with abdominal signs and symptoms, such as ascites, pain, distension, and an abdominal mass [7,25]. They often have a high symptom burden compared to patients with other cancer types. The diagnosis of DMPM may be delayed because of nonspecific symptoms [7,25,26,27]. Thus, many patients with DMPM have advanced disease at presentation [7]. DMPM can spread extensively in the abdomen, but can also and rarely metastasize beyond the abdominal cavity.

The present narrative review aims to present DMPM’s diagnostic and therapeutic pathway.

## 2. Diagnosis and Pathology

Most patients are asymptomatic and diagnosed at an advanced stage, and the median time between symptoms to diagnosis is about four months. The diagnosis of peritoneal mesothelioma could also be established incidentally during abdominal operations in patients with an indolent disease [28].

Cytologic evaluation of serosal effusions has limited diagnostic sensitivity that varies from 30% to 75%, and therefore, it is still not advised for the final diagnosis of DMPM [29]. Moreover, the substantial overlap in atypical characteristics and immunoreactivity between benign reactive and malignant mesothelial cell proliferation poses another challenge for cytological diagnosis. Additionally, the pathologist could be deceived by the fact that the effusion fluid may only contain the underlying reactive epithelioid mesothelial cells, rather than malignant cells in sarcomatoid PM. Exfoliative cytology specimens make it more challenging to make a conclusive diagnosis and highlight the value of close correlation with clinical and imaging findings because they cannot be used to assess invasion of preexisting tissue (not granulation tissue), one of the crucial histologic diagnostic features of DMPM [13]. Additionally, the cytologic examination does not permit the evaluation of the proliferative index using the Ki-67. This prognostic factor is vital to subsequent treatment decision making [30].

The pathological diagnosis of peritoneal mesothelioma should consider pertinent clinical, radiologic, and surgical findings and requires an adequate tissue specimen. The specimen should be obtained by exploratory laparoscopy or core needle biopsy, not by fine needle biopsy of serosal effusion. Most peritoneal mesotheliomas are readily identifiable by routine hematoxylin-eosin staining. There are three major histologic subtypes, categorized as epithelioid, sarcoma, or mixed (biphasic), according to the updated 2015 WHO classification. A definitive diagnosis of DMPM requires workup, including immunohistochemistry (IHC). Positive IHC markers are Calretinin, cytokeratin 5/6, WT-1, podoplanin, and Thrombomodulin. Negative IHC markers are Claudin 4, TTF-1, and CEA [3].

IHC can assist in evaluating BRCA-associated protein 1 (BAP1) expression. BAP1 mutation has been found in about 60% of DMPM [24,31,32,33]. However, its prognostic significance remains to be cleared as some inconsistent results have been published. Leblay (2016) observed that loss of BAP1 nuclear expression and its complete tumor suppressor activity occurred in 57% of their peritoneal mesothelioma cases [34]. This activity loss was underestimated when only the copy number or mutational analyses were considered, suggesting that IHC was more reliable for assessing BAP1 activity. Therefore, integrating different methodologies is advisable to evaluate all nuances of BAP1 gene alterations in malignant mesotheliomas. Offin et al. [35], in contrast to Leblay, presented conflicting results regarding the prognostic significance of the loss of BAP1 nuclear expression or mutation in DMPM. The former claimed a negative association, while the latter a positive association. A different assessment technology, different cohorts’ prognostic profiles, and differences in the therapies can account for such inconsistency between these experiences.

A histological review of a DMPM diagnosis by a pathologist with expertise in Peritoneal Surface Malignancies is mandatory. The pathological report must contain data on the histological subtype (well-differentiated papillary, multicystic, epithelioid, biphasic, and sarcomatoid), invasiveness, necrosis, Ki-67, mitotic rate, and nodal status [26].

## 3. Preoperative Workup

The literature lacks data on the diagnostic performance of imaging assessment of DMPM. The CT scan is currently the preferred radiologic method in the preoperative evaluation of peritoneal mesothelioma [26]. Such popularity may be due to accessibility, cost, and ease of interpretation, even for non-specialized radiologists. Recent data demonstrated that CT scans could assist in the differential diagnosis of peritoneal mesothelioma with other peritoneal surface malignancies [36,37]. A seminal study reported the CT’s clinical usefulness in the preoperative evaluation of DMPM resectability [38]. According to a meta-analysis, the CT scan underestimates the disease burden in the context of PSM.

Magnetic resonance imaging has been suggested to be superior to CT scan in quantifying the PCI in PSM [39]. Recently, fluorine-18 fluorodeoxyglucose (18F-FDG)-PET/contrast-enhanced computed tomography (PET/CT) has become a promising tool for DMPM, with sensitivity, specificity, and accuracy of 86, 89, and 87%, respectively [36]. However, more data are needed to define the actual role and potentialities of PET/CT in the preoperative workup of DMPM.

The clinical utility of baseline serum tumor markers in 60 DMPM patients selected for CRS and HIPEC has been evaluated [37]. Forty-six patients underwent adequate cytoreduction. Baseline diagnostic sensitivities of CA-125, CEA, CA19.9, and CA15.3 were 53.3%, 0%, 3.8%, and 48.5%, respectively.

CA125, using the 35 U/L as a cut-off, was significantly correlated to high-grade histological subtype, PCI > 25, and no preoperative systemic chemotherapy. There are conflicting data on the prognostic significance of baseline serum CA-125 [37,40], but its determination is advisable in the preoperative evaluation.

Circulating mesothelin could be a helpful marker in diagnosing DMPM [41]. In the differential diagnosis of DMPM from other PSM, at a cut-off value of 5.21 ng/dL, mesothelin had sensitivity, specificity, positive predictive value, and negative predictive value of 70%, 100%, 100%, and 61%, respectively. Besides its diagnostic utility, mesothelin represents a valuable druggable target. Amatuximab, a chimeric anti-mesothelin antibody, in combination with cisplatin/Pemetrexed, has provided promising oncological outcomes in unresectable pleural mesothelioma [42] and is currently being tested in a randomized phase II trial in malignant pleural mesothelioma patients [43]

Some studies have explored the clinical utility of laparoscopy in the preoperative evaluation of non-mesothelioma PSMs. The principal advantages are the better assessment of the pre-CRS extent of peritoneal disease, higher accuracy in evaluating disease resectability, and low morbidity and mortality related to the procedure [44,45,46,47]. The procedure allows obtaining adequate specimens for diagnosis and should be done by an expert in PSM. After a throughout evaluation of the peritoneal cavity, the surgeon should calculate the pre-cytoreduction PCI and evaluate the disease resectability, considering the involvement of small bowel serosa and mesentery. Biopsies should not be done in the diaphragmatic peritoneum. Such a maneuver could hamper the resection of this structure in the subsequent operation due to fibrosis and scars.

Diagnostic and therapeutic decision making of DMPM should be conducted by a multidisciplinary team (MDT), which is considered the best practice in cancer and is a critical element of coordinated cancer care [26,48,49].

## 4. Treatment

### 4.1. Systemic Chemotherapy and Immunotherapy

Before the randomized study by Vogelzang et al. (2003), many treatment regimens were available for pleural mesothelioma [50]. According to a review, the four more essential combinations were: cisplatin without doxorubicin, doxorubicin without cisplatin, a combination of cisplatin and doxorubicin, and regimens with neither cisplatin nor doxorubicin. The combination of cisplatin and doxorubicin had the best overall response rate. Combination therapy showed a significantly better response rate than monotherapy (22.6% vs. 11.6%; *p*-value < 0.001).

Following the Vogelzang et al. randomized trial, there was a demand for patient access to Pemetrexed before the approval of the regimen. The introduction of Pemetrexed (Alimta, Eli Lilly), a multitargeted antifolate agent, has improved mesothelioma patients’ outcomes. There was a positive impact on quality of life and survival. Carteni et al. (2008) and Jänne et al. (2005) reported data on unresectable DMPM that received at least one dose of Pemetrexed alone or combined with cisplatin or carboplatin [2,51]. In the Jänne et al. series, 33% of previously treated patients and 21% of chemotherapy-naïve patients received at least 6 treatment cycles. The two series showed higher response rates when Pemetrexed was combined with a platinum agent. Responses up to 30% were obtained with Pemetrexed and cisplatin. Median survival with Pemetrexed alone ranged from 8.7 to 10.3 months.

A multi-institutional phase II study by Simon et al. (2008) tested the combination of Pemetrexed and gemcitabine in 20 unresectable patients [52]. Before the inclusion, 15 patients had at least 1 surgical procedure, and 4 had surgery with curative intent. Fifteen patients completed four or more cycles. One patient died, and another five discontinued treatment due to unacceptable toxicity. A total of 8 patients (40%) experienced grade 4 neutropenia. A total of 2 patients (10%) experienced febrile neutropenia, and 1 (5%) had grade 4 anemia. The disease control rate was 50%, and the median time to progression was 10.4 months. Median OS for all patients was 26.8 months.

Three retrospective studies from Italy (2013), France (2016), and the USA (2018) evaluated the role of perioperative systemic chemotherapy in DMPM treated with CRS and HIPEC [53,54,55]. Generally, these studies have not reported clear criteria for patients’ allocation to one or another group. Pemetrexed combined with a platinum agent was the most frequently used SC regimen.

There was no significant difference between subgroups in terms of overall survival in the Italian and American series. In the French series, the 5-year OS was 40%, 67%, 62%, and 56% in the neoadjuvant, postoperative, and no-chemotherapy groups (*p*-value = 0.049). Neoadjuvant chemotherapy was independently associated with worse oncological outcomes in the Italian series (PFS, HR: 2.47, 95%CI: 1.42–4.29), *p*-value = 0.01) and the French series (OS, HR, 2.30; 95%CI: 1.07–4.94; *p*-value = 0.033). Neither the CC-score at CRS nor grade 3/5 morbidity was linked with preoperative systemic chemotherapy.

According to Deraco et al., the neoadjuvant platinum-Pemetrexed and platinum-gemcitabine combinations resulted in 86% and 82% disease control rates, respectively. The combined median PFS for the two combinations was 14.4 months. While the median OS for platinum and Pemetrexed was not attained, the median OS for platinum and gemcitabine was 31.4 months.

In summary, neoadjuvant systemic chemotherapy can be offered to operable DMPM cases with some concerns regarding disease resectability. As long as the patient has good clinical conditions (ECOG PS3), platinum-based systemic chemotherapy with palliative purpose should be offered in non-operable and/or unresectable DMPM patients, rather than the best supportive care. (Figure 1) In cases where at least one poor prognostic marker is present (CC-score > 1, sarcomatoid or biphasic subtype, lymph node involvement, Ki67 > 9%, or PCI > 17), patients should receive adjuvant combination systemic chemotherapy. If not, the patient might merely receive follow-up care. The best suggested treatment plan is cisplatin and Pemetrexed; the second option is the cisplatin and gemcitabine combination (Figure 1) [26].

Immune checkpoint inhibitors (ICIs) have recently gained enormous popularity in treating various types of cancer. Although nivolumab and ipilimumab are available to patients with pleural mesothelioma as a standard first-line option, based on the results of Checkmate-743 and MAPS trials, their use in DMPM upfront is still contentious. Until recently, pleural mesothelioma and DMPM were considered dissimilar in their expression of PD-L1 (nearly 50% in DMPM vs. 30% in pleural mesothelioma) [56]. However, different results have been reported in a recent large-scale molecular analysis study involving 1294 pleural mesothelioma (n = 980) and DMPM (n = 314) patients [57]. It was shown that the tumor’s location was not associated with different expressions of PD-L1 (approximately 50% expression for both pleural mesothelioma and DMPM). It is well known that tumors with high microsatellite instability (MSI-H) are hypermutated and produce many peptides that function as neoantigens, causing a robust immune response characterized by a high number of tumor-infiltrating lymphocytes. The occurrence of MSI-H in mesotheliomas has been shown to be extremely rare; it was present in just 1 case (out of 1.294) in the Dagogo cohort [57]. Moreover, DMPM had a lower prevalence of tumor mutational burden than pleural mesothelioma (median TMB 1.25 mut/Mb in DMPM v 1.74 mut/Mb in pleural, *p*-value: 1.89 × 10^−3^) [57].

Clinical trials on pleural mesothelioma have tended to exclude DMPM from the eligibility criteria due to its rare incidence. Therefore, the extrapolation of recommendations on ICI use in pleural mesothelioma to DMPM is cumbersome. Clinical studies testing ICIs specifically in DMPM are usually of a very low level of evidence, but some series have reported some promising results. Twenty advanced yet previously treated DMPM cases were included in a second phase II study that examined the use of atezolizumab in combination with bevacizumab. The overall response rate was 40%, and the 1-year PFS and 1-year OS were both remarkable: 61% and 85%, respectively [58]. The higher the epithelial-mesenchymal transition gene scores were, the shorter the progression-free survival on atezolizumab/bevacizumab and prior platinum Pemetrexed chemotherapy. In another clinical trial, 29 DMPM cases were included, of which 20 were treated with dual ICIs (nivolumab plus ipilimumab) and 9 with a single-agent ICI [59]. No difference between the single and two-agent ICIs regarding overall response rate was found.

### 4.2. Local-Regional Treatment

CRS and HIPEC are currently the standard of care in selected patients with DMPM, with promising results in terms of median overall survival after the operation that ranges from 34 to 92 months [11,12,60,61,62]. Data reporting outcomes of patients treated with CRS and HIPEC are derived from single-center institutional reviews, two large multicenter reviews, and a recent meta-analysis.

#### 4.2.1. Eligibility for Cytoreductive Surgery (CRS) and HIPEC

##### Evaluation of Operability and Disease Resectability

Criteria for operability are well-standardized and universally accepted [63]. They could be summarized as age <75; Eastern Cooperative Oncology Group (ECOG) performance score <2; no relevant comorbidities; and no extra-abdominal metastases and resectable peritoneal disease at preoperative CT-scan.

The evaluation of disease resectability using the CT scan was first addressed by Yan et al. [38], who analyzed the preoperative CT scans of DMPM patients treated with CRS and HIPEC. Patients were categorized into two groups according to residual disease after CRS (CC0/1 vs. CC2/3). A total of 7 patients (64%) in the incomplete cytoreduction group and 2 patients (11%) in the complete cytoreduction group had a >5 cm tumor mass in the epigastric region (*p*-value = 0.004). Moreover, CT scans showed architectural distortion of small bowel mesentery in 9 patients (82%) in the incomplete CRS group and 2 patients (11%) in the complete cytoreduction group (*p*-value < 0.001). The simultaneous presence of >5 cm tumor mass in the epigastric region and the loss of mesenteric architecture was associated with a nil probability of complete CRS, while the absence of the 2 CT features was associated with a 94% probability of complete CRS.

Recently, Sugarbaker et al. described CT features specifically for malignant peritoneal mesothelioma, which are significantly associated with worse survival. A total of 100 patients with a preoperative CT were evaluated in terms of 11 concerning CT features. In total, 9 of the 11 CT features statistically significantly predicted reduced survival in those patients treated with CRS and HIPEC, which could be of great value in preoperative decision making (Figure 2) [64].

Laterza et al. reported using preoperative laparoscopy in 33 DMPM patients who underwent CRS and HIPEC [65]. Regarding the specific disease sites involved, neither laparoscopic nor surgical exploration revealed any case with epigastric lesions larger than 5 cm in diameter. The small bowel and associated mesentery involvement were seen during laparoscopy in three patients, but it was confirmed during surgical investigation in four cases. Laparoscopy accurately predicted resectability with 100% sensitivity, 75% specificity, 97% positive predictive value, and 100% negative predictive value.

##### Prognostic Factors and Risk Prediction Models

The process of selecting patients requires a well-thought-out interpretation of prognostic profiles. In DMPM, prognostic factors have been reported by several authors. Age, histological subtype, complete cytoreduction, and disease extent are the most well known [12,60,66,67]. The Ki-67 proliferative index has recently emerged as a reliable prognostic factor. The expression of the PD-L1 level has also been suggested as a good candidate predictor [68]. However, the current literature lacks tools to provide a person-specific survival prediction in DMPM. Yan et al., in a review of 294 DMPM patients undergoing CRS/HIPEC, proposed a tumor, node, and metastasis staging system. This system is limited for preoperative prognostic evaluation, as it relies on lymph node status, which can only be determined after surgery [69].

Using machine-learned Bayesian belief networks with stepwise training, testing, and cross-validation, Schaub et al. created a preoperative nomogram that predicts survival in DMPM. The nomogram is based on the preoperative serum CA-125, pre-CRS PCI, and histological subtype [40]. This nomogram has a good discriminative capacity, as the mean area under ROC of the 3- and 5-year models were 0.77 and 0.74, respectively.

Another prognostic tool was developed using the conditional inference tree model [30]. This user-friendly model provides a simple-to-understand graphic output that can help in the patient selection for CRS and HIPEC. Pre-cytoreduction PCI and the Ki-67 proliferative tumor index are the two critical parameters of this decision-tree model that discriminates three subgroups: (I) Ki-67 ≤ 9%; (II) Ki-67 > 9% and PCI ≤ 17; and (III) Ki-67 > 9% and PCI > 17. In subgroups I, II, and III, the median OS was 86.6, 63.2, and 10.3 months, respectively. The model helps to identify a subset with a poor prognosis (III) that would not benefit from CRS and HIPEC. It also has an adequate discriminant capacity with a bootstrap-corrected Harrel c-index of 0.74.

##### Biphasic/Sarcomatoid Histologies

Biphasic mesothelioma represents a distinct and rare subtype that has traditionally been grouped with the sarcomatoid variant and analyzed separately from the epithelioid counterpart. Given the highly dismal prognosis of the sarcomatoid and biphasic subtypes, they have been deemed contraindications for CRS/HIPEC. To clarify the actual outcome of biphasic peritoneal mesotheliomas after complete CRS and HIPEC, data from an International Registry on Peritoneal Mesothelioma were analyzed. From a cohort comprising 484 DMPM cases treated with complete CRS and HIPEC, the authors selected 34 biphasic cases for the study. For patients with CC0 resection, the median year OSs were 7.8 and 6.8 years (*p* = 0.015) for epithelioid and biphasic mesotheliomas, respectively. Including CC1 resections in the analysis resulted in inferior median OSs of 7.8 and 2.8 years (*p* = 0.0012), respectively [70]. The authors concluded that long-term survival is achievable in highly selected cases and that biphasic histology should not be considered an absolute contraindication for CRS/HIPEC if there is a low-volume disease and if complete cytoreduction can be achieved.

In summary, absolute contra-indications to CRS and HIPEC are sarcomatoid histology, massive pleural invasion, massive infiltration of small bowel serosa, involvement of cardiophrenic angle lymph nodes, and bulky retroperitoneal lymph nodes. In contrast, the relative contra-indications are biphasic histology, invasion of small bowel mesentery, disease not amenable to cytoreduction down to CC0/1, Ki-67 > 67, PCI > 17, and massive diaphragmatic muscle involvement [26].

#### 4.2.2. Technical Aspects of Cytoreductive Surgery

Cytoreductive surgery is a standardized surgical strategy that comprises a systematically ordered sequence of surgical procedures. As the level of surgical effort should logically be modulated according to the biological aggressiveness of the tumor, CRS could have a technical variation depending on the type of PSM. The Milan Center has suggested a more radical cytoreduction with a whole parietal peritonectomy resection in DMPM [71].

Such a recommendation has been criticized, as the peritoneal layer is regarded as the first oncological defense line. Resecting the parietal peritoneum completely in every case might theoretically favor tumor progression, as it might hamper the patients’ immunological response to the disease [72]. Another criticism of complete parietal peritonectomy is that the parietal peritoneum accounts for only 18% of the total peritoneal area (viscera and parietal combined) [1].

A matched controlled study of DMPM compared complete versus partial parietal peritonectomy during CRS. There was a prognostic advantage in favor of the complete parietal peritoneal resection group. In multivariate analysis, the type of peritoneal resection was an independent prognostic factor, along with complete CRS, node negativity, epithelioid histology, and lower MIB-1 proliferating index. Morbidity and reoperation rates did not differ between groups. There were no surgical deaths. In 12 of the 24 patients who underwent parietal peritonectomy, pathologic examination revealed disease involvement on the parietal surface, while there was no apparent macroscopic tumor during the operation [71].

The retroperitoneal lymph node status assessment is not carried out consistently and systematically across international PSM centers, despite its prognostic importance being acknowledged in the most relevant DMPM cohorts.

Only the epithelial subtype, lack of lymph node metastases, completion of cytoreduction, and use of HIPEC were independently linked with improved outcomes in multivariate analysis from the multi-institutional data registry containing 405 DMPM cases [12].

The Washington Cancer Center has reported similar findings. All 7 out of 100 DMPM patients that were positive for lymph nodes died 2 years after the operation. A total of 50% of the 93 patients still alive after 5 years survived. Female gender, lymph node metastases, epithelial type, and complete cytoreduction were all found to be independently linked with better survival [73].

In the Milan experience, from 83 patients with DMPM submitted to surgical cytoreduction and HIPEC, 38 patients had their lymph nodes evaluated; 11 had positive lymph nodes, whereas 27 had negative results. In 45 patients, lymph nodes were not clinically suspicious and were not resected during the operation. The most frequently affected nodes were the paracolic (n = 2) and iliac (n = 7). Pathologically negative nodes (vs. positive/not assessed) were independently associated with longer OS in multivariate analysis (hazard ratio (HR) = 2.81; 95%CI = 1.12 to 7.05; *p*-value = 0.027) [74].

Carefully sampling the non-suspicious retroperitoneal lymph nodes and resecting suspicious ones are advisable [26].

In peritoneal surface malignancies with intermediate biological aggressiveness, such as epithelioid DMPM, small- to medium-sized nodules and plaques are frequently present on the mesentery surface, with minimal deep tissue invasion. In such a situation, both sides of the mesentery may undergo a peritonectomy [75,76]. The serosal layer is removed 3–4 cm distally from the bowel boundaries. Any vascular trauma must be avoided, especially near the small bowel, where the terminal arteries are fragile. Vascular damage may cause insufficient blood supply and necessitate further intestine resection. This procedure is performed by identifying the cleavage plain between the serosal layer and the mesenteric fat tissue. The maneuver continues by stripping and using the electrosurgical device or blunt dissection. Isolated minor disease localizations can be electro-evaporated.

#### 4.2.3. The Role of HIPEC and HIPEC Drug Schedules

According to 2 large multicentric observational studies, HIPEC following surgery was independently associated with better survival, with a reduction of risk of death ranging from 46% to 53% [12,66]. Only the study by Yan TD specified the HIPEC protocols, and the most frequent regimens were cisplatin and doxorubicin, cisplatin and mitomycin-C, cisplatin, mitomycin-C, or paclitaxel. Despite the adequate control of confounders by multivariate analysis, it is impossible to rule out a selection bias. Considering the very low prevalence of the disease and the consequent difficulty in conducting RCT, one would reasonably expect that future studies are unlikely to provide a more precise magnitude of treatment effect.

The multicentric French registry (RENAPE) was retrospectively evaluated to assess the prognostic impact of different drugs for HIPEC in 249 DMPM patients submitted to CRS [77]. The drug combinations were cisplatin alone (n = 21), cisplatin and doxorubicin (n = 60), cisplatin and mitomycin-C (n = 52), mitomycin-C alone (n = 15), oxaliplatin alone (n = 52), and oxaliplatin and irinotecan (n = 49). Patients receiving dual-drug HIPEC had a significantly better OS when compared to mono-drug HIPEC [HR: 0.54 (95%CI 0.31–0.95), *p* = 0.03]. There was no increase in severe postoperative morbidity risk [OR 0.86 (95%CI: 0.36–1.11), *p* = NS].

In a retrospective comparative analysis of 211 DMPM patients from 3 referral centers in the US, the authors observed that the use of cisplatin with sodium thiosulfate (STS) was independently associated with prolonged survival, compared to mitomycin-C [HR: 0.58 (CI 95%: 0.38–0.91), *p*-value: 0.01] [60].

A small observational study comparing HIPEC with carboplatin (600–800 mg/m²) vs. mitomycin-C (30–40 mg fixed dose) after CRS in DMPM patients showed a survival advantage for carboplatin, without a difference in terms of mortality [78]. No multivariate analysis was conducted to assess prognostic factors. Still, there was no significant difference between the groups regarding age, gender, ECOG performance status, histological subtype, and surgery radicality.

In summary, the HIPEC regimen should be cisplatin-based and preferably associated with a second drug to maximize the cytotoxic effect and optimize the survival benefit [33].

#### 4.2.4. Normothermic Intraperitoneal Chemotherapies

Sugarbaker et al. proposed the addition of postoperative intraperitoneal chemotherapies—i.e., early postoperative intraperitoneal chemotherapy (EPIC) and non-hyperthermic intraperitoneal chemotherapy (NIPEC)—to CRS and HIPEC [79].

Multiple studies have reported on EPIC employment. The lack of characterization and uniformity of the chemotherapeutic agents used, the number of days, and the criteria for EPIC indication, combined with the small number of patients who receive therapy, do not allow for consistent conclusions.

Recently, EPIC and NIPEC were compared in 129 epithelioid DMPM patients after excluding low-grade and poorly differentiated tumors [80]. Three groups were compared: CRS-HIPEC, CRS-HIPEC-EPIC, and CRS-HIPEC-EPIC-NIPEC. HIPEC was performed using cisplatin/doxorubicin, EPIC with paclitaxel, and NIPEC with paclitaxel or Pemetrexed. There was significantly better survival in the NIPEC group (*p*-value = 0.037). An evaluation of patients with/without NIPEC confirmed a significant survival difference (*p*-value = 0.011). The addition of EPIC to HIPEC did not translate into improved outcomes. However, statistically significantly better survival was seen when multiple cycles of NIPEC were utilized. This loss of distinction with the addition of EPIC over time, with marked benefit associated with repeated cycles of local chemotherapy, suggests the potential benefit of long-term IP-directed treatment.

In a small phase II study, bidirectional chemotherapy with IP Pemetrexed combined with IV cisplatin was performed after CRS and HIPEC ± EPIC. Eight patients were epithelioid and two biphasic; four were CC-0/1, four were CC-2, and two were CC3. In total, 9 of 10 patients completed all 6 cycles of therapy without treatment delays or dose modifications. One patient developed a catheter infection after cycle three and required catheter removal. The median survival for all 10 patients was 33.5 months [81]

More recently, a French group used pressurized intraperitoneal aerosol chemotherapy (PIPAC) with 26 cases of unresectable DMPM [82]. Overall, 79 PIPAC procedures were performed, with half of the patients receiving > 3 PIPAC procedures. Among 8 patients (31%), 10 adverse events (13% of procedures) were reported, including 2 severe complications. Improvement of symptoms occurred in 32% of the patients, whereas control of ascites was reported in 46%. All but 1 procedure among 14 patients (54%) secondarily treated by CRS-HIPEC were considered complete resections. After a median follow-up period of 29.6 months, the median OS was 12 months. These outcomes are promising, especially the conversion rate, but more evidence data is needed.

## 5. Follow-Up of DMPM

The post-treatment follow-up program should aim to diagnose potentially resectable recurrences and to continuously evaluate early and long-term treatment-related sequelae. Available data on this topic are scarce; therefore, it is challenging to define the optimal follow-up program. DMPM patients usually have a median progression-free survival between 13.9 and 25.1 months [40,54,66]. However, if the recurrence is unresectable, there is no standardized second-line treatment option, as DMPM is a chemo-resistant disease [54,60,62].

Since almost 70% of relapses occur in the first 2 years after treatment, follow-up should be done with more frequent assessments in the first 24 months. Another important aspect is the duration of supervision. Baratti et al. reported 108 patients with DMPM who underwent complete CRS-HIPEC with cisplatin and doxorubicin or mitomycin-C [68]. After a median follow-up of 48.8 months, the 5- and 10-year PFSs were 38.4% and 35.9%, respectively. The survival curve reached a plateau after seven years. Among the 19 long-term survivors of over 7 years, the median survival was 104.2 months (95%CI: 91.4 to 133.6). A large national registry [67] reported on 1514 patients, dividing them into 5 groups: observation (25%), chemotherapy alone (24%), CRS alone (13%), CRS/chemotherapy (23%), and CRS-HIPEC (14%). During a median follow-up of 50 months, the median survival for CRS and HIPEC was 61 months. As with Baratti’s data, the number of deaths decreased steadily after about 85 months of follow-up. Although DMPM may have a high tendency to remain in the peritoneal cavity for most of its natural history, some cases recur outside the peritoneal cavity during post-treatment follow-up.

Baratti et al. found that almost 18.4% of cases had treatment failure outside the abdominal cavity and involved the pleural and retroperitoneal lymph nodes. Therefore, in addition to the abdominal cavity, the chest should also be considered in the following evaluation [62]. Follow-up of DMPM patients during the first two years and then every six months after CRS-HIPEC is recommended and includes every six months: a physical examination, tomographic scan of the chest/abdominal/pelvic region, and dosing of the CA125 biomarker. The advisable duration of follow-up is seven years after the surgery, in contrast to five years for other peritoneal metastatic diseases, such as colon cancer [26].

## 6. Borderline Histological Subtypes

MCPM accounts for 3–5% of peritoneal mesothelioma and primarily affects women of reproductive age [83]. Diagnosis is usually concomitant or secondary to nonspecific abdominal symptoms [84]. Typical MCPM histology shows borderline signs, with an absence of cell atypia or increased mitotic count. However, squamous metaplasia is possible [85]. The lesions consist of small cysts with benign mesothelium that appear as cuboidal cells that occasionally form papillae. The intercystic stroma is characterized by varying degrees of inflammation [86]. Pathologic differential diagnosis includes many benign and malignant lesions presenting as polycystic masses in the abdomen, usually revealed by immunohistochemistry [83,87]. The unique epidemiological features of a MCPM patient compared with those of a DMPM patient (young female with no history of asbestos exposure) suggest an independent etiology. However, hormonal or repetitive peritoneal irritation hypotheses have not been definitively established [85,86,88,89].

Well-differentiated papillary peritoneal mesothelioma (WDPPM) is a morphologically distinctive papillary proliferation of mesothelial cells that is most commonly identified as an incidental finding in the peritoneal cavity. These lesions may be single or multiple, but by definition, they do not invade the underlying stroma, are usually benign or indolent, and can sometimes last for years [90]. However, the nature of WDPPM is controversial, with various theories ranging from reactive non-neoplastic processes to benign tumors to epithelial malignant mesothelioma variants or precursors [91]. Even more confusing is that DMPMs may have areas that mimic WDPPM. Because DMPM is an aggressive tumor, it is essential to differentiate it from WDPPM. Three features of MCPM and WDPM should guide treatment strategies: a high recurrence rate after surgery is estimated to occur in up to 50% [83,92]; a rare potential for malignant transformation despite indolent behavior, especially in the case of WDPM; and concerns regarding fertility capacity in the case of women of reproductive age. CRS and HIPEC have been reported to be associated with the best long-term outcomes in both histologies [26].

## 7. Conclusions

DMPM is a very rare disease with a poor prognosis. Cytoreductive surgery and HIPEC remain the cornerstone of the treatment, and their success depends on proper patient selection for surgery. The therapeutic decision making requires the wise and judicious analysis of prognostic factors. DMPM should be managed in a specialized referral center located in a centralized context due to the extremely low incidence of this clinical entity.

## Figures and Tables

**Figure 1 cancers-15-00662-f001:**
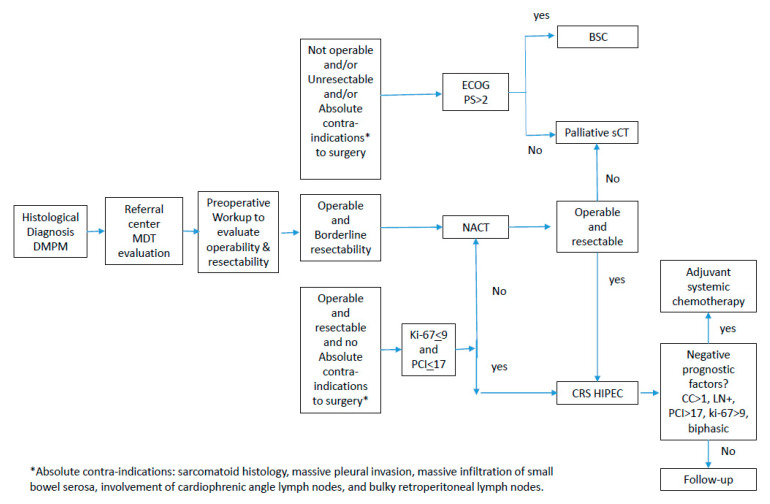
Diagnostic and therapeutic pathway in diffuse malignant peritoneal mesothelioma.

**Figure 2 cancers-15-00662-f002:**
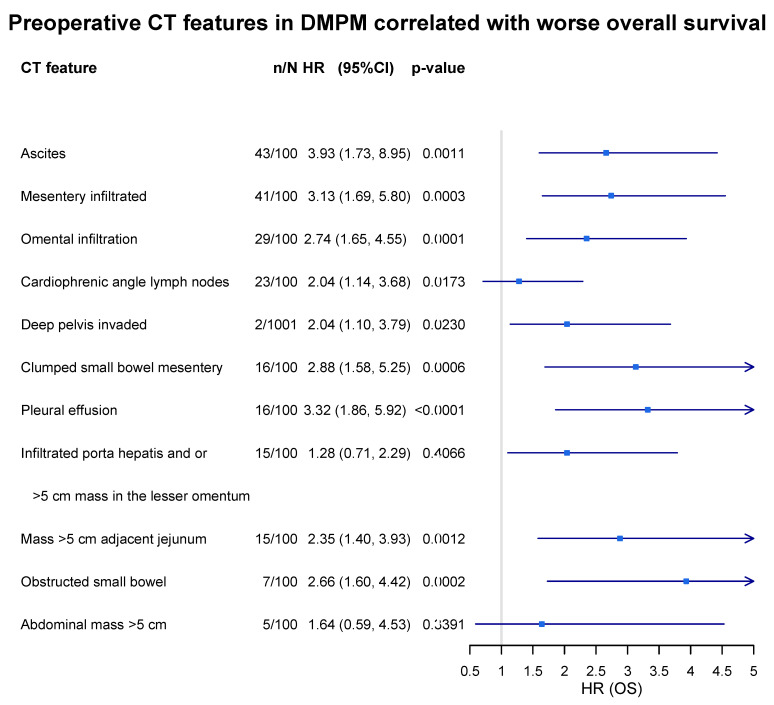
Prognostic significance of radiological features evaluated by preoperative chest abdominal CT scan in DMPM patients. n/N: incidence of feature in 100 DMPM cases.

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
