# Peer review of "Diagnostic and Therapeutic Pathway in Diffuse Malignant Peritoneal Mesothelioma"

_cancers, 2023, doi:10.3390/cancers15030662_

Round 1

Reviewer 1 Report

Overall, well written and comprehensive review of a rare disease. 

1) Given the heterogenous survival of patients depending on histology and treatment received the 1 year OS rate of 46% seems a bit misleading. Would emphasize the heterogeneity of response, indolence etc. 

2) 'Many patients with DMPM have an idiopathic disease' - please clarify. 

3) Would consider mentioning short latency period for asbestos exposure to peritoneal meso. 

4) Would suggest remaining uniform in the abbreviation used. Prefer DMPM to just PM. 

5) Clarify association of BAP1 alterations and IHC. I assume the 2 studies you are presenting are contradicting. Any reason they may be different? Different patient populations? Different type of sequencing used?

6) The authors could consider addressing papillary histology or benign multicystic peritoneal mesothelioma since these follow a slightly different paradigm. 

Author Response

Many thanks to the reviewer for the precious comments and suggestions. The paper was modified accordingly and the changes were highlighted in yellow.

Reviewer 1

Overall, well written and comprehensive review of a rare disease. 

1) Given the heterogenous survival of patients depending on histology and treatment received the 1-year OS rate of 46% seems a bit misleading. Would emphasize the heterogeneity of response, indolence etc. 

Response: The cited data on survival were estimated in a population-based study from SEER database, where 1998 patients, encompassing different histological subtypes and submitted to different types of therapy, were analyzed. The information was better presented by adding the SD, to the provided survival estimates.

2) Many patients with DMPM have an idiopathic disease' - please clarify.

Response: "idiopathic", in the context of the paragraph, meant that the disease was caused by factors other than asbestos exposure. As this information is presented again in the next subsequent phrases (about 33–50% of patients diagnosed with DMPM report any known prior exposure to asbestos), to avoid redundancy, the string was omitted.

3) Would consider mentioning short latency period for asbestos exposure to peritoneal meso.

Response: we would prefer not to report such data in our review as there is no clear-cut literature on whether the asbestos post-exposure latency depends on the location. According to Marinacci A et al. (2007), there was no correlation (median latency 44-45 years for pleural mesothelioma and 36-42 for peritoneal). According to D'Agostin F et al. (2017) 48.7+/-10.8 for pleural and 49.9+/-10.5 for peritoneal.

4) Would suggest remaining uniform in the abbreviation used. Prefer DMPM to just PM.

Response:  Many thanks. We corrected the paper accordingly.

5) Clarify the association of BAP1 alterations and IHC. I assume the 2 studies you are presenting are contradicting. Any reason they may be different? Different patient populations? Different types of sequencing used?

Response: Leblay (2016) observed that loss of BAP1 nuclear expression and its complete tumor suppressor activity occurred in 57% of their peritoneal mesothelioma cases. This activity loss was underestimated when only copy number or mutational analyses were considered, suggesting that IHC was more reliable for assessing BAP1 activity. Therefore, integrating different methodologies is advisable to evaluate all nuances of BAP1 gene alterations in malignant mesotheliomas. Offin and Leblay presented contrasting results regarding the prognostic significance of the loss of BAP1 nuclear expression or mutation in DMPMs. The former claimed a negative association, while the latter was positive. There are some reasons to explain such a conundrum. The technologies assessing BAP1 loss of expression/mutation were different; both series were different regarding prognostic profiles and treatments. Offin reported on DMPM cases, with nearly one-third of cases with extra-abdominal diseases. In contrast to the Leblay cohort, they were not uniformly submitted to CRS and HIPEC, which is considered the current standard of care.

6) The authors could consider addressing papillary histology or benign multicystic peritoneal mesothelioma since these follow a slightly different paradigm.

Response: we added another section to the paper containing information regarding these borderline histologies.

Reviewer 2 Report

It’s an interesting review. The authors described the diagnosis and treatment pathways of the rare cancer, diffuse malignant peritoneal mesothelioma. The review indicated the difficulty in early diagnosis and the limitation of the treatment. The detailed local-regional treatment pathways were described in the review. I have some major comments as follows:

1 The abstract need to re-write, as the abstract is almost a part of introduction. The authors should include the contents they mainly described in the review. The abstract should be mainly a summary of what mentioned in the review, not just introduction plus one sentence to summarize.

2 In the part 4.1 systemic chemotherapy and immunotherapy, it will be better to add the date, when the authors mentioned the clinical trials, like which year.

3 In part 4.1 systemic chemotherapy and immunotherapy, the last paragraph, the authors mentioned that ICIs become the second-line treatment for DMPM patients. Why does ICIs become second-line treatment? As far as I know, ICIs become the first-line treatment for malignant pleural mesothelioma (MPM). It will be very interesting if the authors explain why ICIc treatment for DMPM is different from MPM in the review.

4 In the table 1, one of the CT features is pleural effusion. Why is pleural effusion one CT feature? Will DMPM induce pleural effusion in some patients?

Author Response

Many thanks to the reviewer for the precious comments and suggestions. The paper was modified accordingly and the changes were highlighted in yellow.

Reviewer 2

It's an interesting review. The authors described the diagnosis and treatment pathways of the rare cancer, diffuse malignant peritoneal mesothelioma. The review indicated the difficulty in early diagnosis and the limitation of the treatment. The detailed local-regional treatment pathways were described in the review. I have some major comments as follows:

1 The abstract need to re-write, as the abstract is almost a part of introduction. The authors should include the contents they mainly described in the review. The abstract should be mainly a summary of what mentioned in the review, not just introduction plus one sentence to summarize.

Response: the abstract was rewritten 

2 In the part 4.1 systemic chemotherapy and immunotherapy, it will be better to add the date, when the authors mentioned the clinical trials, like which year.

Response: the year of the studies were added.

3 In part 4.1 systemic chemotherapy and immunotherapy, the last paragraph, the authors mentioned that ICIs become the second-line treatment for DMPM patients. Why does ICIs become second-line treatment? As far as I know, ICIs become the first-line treatment for malignant pleural mesothelioma (MPM). It will be very interesting if the authors explain why ICIc treatment for DMPM is different from MPM in the review.

Response: Although nivolumab, ipilimumab are available to patients with pleural mesothelioma as a standard first-line option, based on results of Checkmate-743 and MAPS trials, their use in DMPM in upfront is still contentious. Untill recently pleural and DMPM thought to be dissimilar in their expression of PD-L1 (nearly 50% in DMPM vs. 30% in pleural mesothelioma)(Chapel DB et al. 2019). However, in a recent large scale molecular analysis study involving 1.294 pleural mesothelioma (n=980) and DMPMs (n=314) different results have been reported. It was shown that the location of the tumor was not associated with different expression of PD-L1 (+/- 50% in both pleural and DMPM). (Dagogo JI et al. 2021) The occurrence of MSI-H in mesotheliomas has been shown to be extremely rare; it was present in just one case (out of 1.294) in a Dagogo’s cohort. Moreover, DMPM had a lower prevalence of tumor mutational burden as compared to pleural mesothelioma (median TMB 1.25 mut/Mb in DMPM v 1.74 mut/Mb in pleural, p-value: 1.89E-03) (Dagogo JI et al. 2021). Clinical trials on pleural mesothelioma have tended to exclude DMPM from the eligibility criteria due to its rare incidence. Therefore, the extrapolation of recommendations on ICI use in pleural mesothelioma to DMPM is cumbersome. Studies testing ICIs specifically in DMPM are usually of low very level of evidence, but some promising results have been reported by some series.

4 In table 1, one of the CT features is pleural effusion. Why is pleural effusion one CT feature? Will DMPM induce pleural effusion in some patients?

Response: yes, DMPM can induce pleural effusion in some cases. In Sugarbaker's experience (Sugarbaker, Cahng and Jelinek, EJSO, 2021), such finding was present in 16% of DMPM patients and was correlated with poorer survival (HR: 3.32). The fact that 15/16 died of peritoneal mesothelioma suggests they were not bi-district diseases (pleural + peritoneal). The pathophysiology of such a phenomenon is still unclear but is could to be related to invasion from the affected contiguous diaphragm, migration of tumor cells from the peritoneal cavity through lymphatic communications, or hematogeneous spread.

Round 2

Reviewer 2 Report

Comments were addressed. Try to make Figure 2 nicer, by adjusting space, and the lines of HR(OS).

Author Response

Many thanks, the graph was modified as suggested
